# Identifying Effects of Urinary Metals on Type 2 Diabetes in U.S. Adults: Cross-Sectional Analysis of National Health and Nutrition Examination Survey 2011–2016

**DOI:** 10.3390/nu14081552

**Published:** 2022-04-08

**Authors:** Jingli Yang, Kayue Chan, Cheukling Choi, Aimin Yang, Kenneth Lo

**Affiliations:** 1College of Earth and Environmental Sciences, Lanzhou University, Lanzhou 730000, China; yangjl18@outlook.com; 2Department of Applied Biology and Chemical Technology, The Hong Kong Polytechnic University, Hong Kong SAR, China; kellycky.chan@connect.polyu.hk (K.C.); cheuk-ling-fion.choi@connect.polyu.hk (C.C.); 3Department of Medicine and Therapeutics, The Chinese University of Hong Kong, Prince of Wales Hospital, Hong Kong SAR, China; 4Hong Kong Institute of Diabetes and Obesity, The Chinese University of Hong Kong, Hong Kong SAR, China; 5Research Institute for Smart Ageing, The Hong Kong Polytechnic University, Hong Kong SAR, China

**Keywords:** urinary metals, diabetes, cardiometabolic disease, Bayesian kernel machine regression

## Abstract

Growing evidence supports the associations of metal exposures with risk of type 2 diabetes (T2D), but the methodological limitations overlook the complexity of relationships within the metal mixtures. We identified and estimated the single and combined effects of urinary metals and their interactions with prevalence of T2D among 3078 participants in the NHANES 2011–2016. We analyzed 15 urinary metals and identified eight metals by elastic-net regression model for further analysis of the prevalence of T2D. Bayesian kernel machine regression and the weighted quantile sum (WQS) regression models identified four metals that had greater importance in T2D, namely cobalt (Co), tin (Sn), uranium (U) and strontium (Sr). The overall OR of T2D was 1.05 (95% CI: 1.01–1.08) for the positive effects and 1.00 (95% CI: 0.98–1.02) for the negative effect in the WQS models. We observed positive (*P*_overall_ = 0.008 and *P*_non-linear_ = 0.100 for Co, *P*_overall_ = 0.011 and *P*_non-linear_ = 0.138 for Sn) and inverse (*P*_overall_ = 0.001, *P*_non-linear_ = 0.209 for Sr) linear dose–response relationships with T2D by restricted cubic spline analysis. Both additive and multiplicative interactions were found in urinary Sn and Sr. In conclusion, urinary Co, Sn, U and Sr played important roles in the development of T2D. The levels of Sn might modify the effect of Sr on T2D risk.

## 1. Introduction

The global diabetes prevalence in 2021 was estimated to be 10.5%, with 537 million people affected by diabetes, 90% of whom had type 2 diabetes (T2D) [1]. Growing numbers of epidemiological studies and evidence support the link between environmental pollution such as heavy metals and the development of diabetes and cardiovascular disease [2,3,4,5]. Compared with some well-established risk factors for diabetes, such as genetic composition, lifestyle habits, and social determinants [6,7], the impact of environmental chemicals and especially metal mixtures on diabetes development has presented challenges in terms of evaluating their overall and specific effects as well as joint effects that might be both detrimental or beneficial [8].

Heavy metals are widely present in the environment, and most people are exposed to them over their lifetimes. Previous epidemiological studies exploring the associations between single metal exposure and risk of T2D have reported inconsistent results [3,9,10,11,12,13]. For example, individuals with increased urinary manganese (Mn) and lead (Pb) had an increased sex-specific risk of hyperglycemia and metabolic syndrome [3,9], while higher urinary cadmium (Cd) levels had no significant association with T2D [12,13]. These divergent results call for further study with advanced statistical methods to identify the main effects and specific effects of metal mixtures on diabetes. However, most previous studies have estimated the association of single-metal exposure with T2D risk by adjusting for other multiple metals using the traditional regression models simultaneously, or introducing cross-product terms [14,15], which may overlook the complexity of relationships within metal mixtures [16,17].

Therefore, we systematically identified and estimated the main- and single-metal effects of urinary metal mixtures as well as their interactions with the prevalence of T2D in the United States (U.S.) National Health and Nutrition Examination Survey (NHANES 2011–2016).

## 2. Materials and Methods

### 2.1. Study Population

The NHANES is a program designed to assess the health and nutritional status of adults and children (≥6 years old) in the U.S. [18]. The survey is conducted by the National Center for Health Statistics (NCHS) and combines data collected from interviews and physical examinations. Since manganese (Mn), strontium (Sr), and tin (Sn) were not detected until 2011, we limited our study to participants with data on urinary metals in three NHANES surveys between 2011 and 2016. Among 29,902 participants, we excluded those aged <18 years (*n* = 11,933), without metal measurements (*n* = 12,384), and missing covariates (*n* = 2507) (Appendix A). We included 3078 participants in the final analysis. All participants gave written informed consent. Ethical approval was obtained from the National Center for Health Statistics Ethics Review Board (Protocol Number: 2011-17).

### 2.2. T2D Definition

The following criteria were used to identify participants with T2D [19]: (1) fasting plasma glucose concentrations ≥ 126 mg/dL (7.0 mmol/L), or (2) glycohemoglobin A1c (HbA1c) levels ≥ 6.5%, or (3) the 2 h plasma glucose ≥ 200 mg/dL (11.1 mmol/L) during a 75 g oral glucose tolerance test, or (4) self-reported diabetes or taking insulin and/or oral glucose-lowering drugs.

### 2.3. Measurement of Urinary Metal Concentrations

The 15 urinary metal levels in this study were measured by inductively coupled plasma mass spectrometers (ELAN^®^ 6100 DRC^Plus^ or ELAN^®^ DRC II, PerkinElmer Norwalk, Fairfield) in three NHANES survey cycles between 2011 and 2016 and included antimony (Sb), arsenic (As), barium (Ba), cadmium (Cd), cesium (Cs), cobalt (Co), lead (Pb), manganese (Mn), mercury (Hg), molybdenum (Mo), strontium (Sr), thallium (TL), tin (Sn), tungsten (W) and uranium (U). Spot urine was collected at the time of the laboratory exam, shipped on dry ice to the National Center for Environmental Health (NCEH) in Atlanta, GA, and stored frozen at −20 °C until assayed [20]. Urinary concentrations of 15 metals below the limits of detection (LODs) were imputed with LOD divided by the square root of 2 [21] (Appendix A).

### 2.4. Covariates

Standardized questionnaires were used to collect socio-demographic, lifestyle, clinical, and nutritional risk factors from participants in the NHANES. We selected potential risk factors based on a priori knowledge of the risk of diabetes [22,23], including sex (men or women), age (18–39, 40–59 or ≥60 years), race/ethnicity (Non-Hispanic White, Non-Hispanic Black, other Hispanic, or other race), educational attainment (less than high school, high school, or at least some college), poverty-income ratio (PIR, <1 or ≥1), smoking status (never, former smoker, current smoker), alcohol consumption (yes or no), physical activity (substandard, standard), body mass index (BMI, ≤25, 25.1–29.9 or ≥30 kg/m^2^), average daily energy intake (<1515, 1515–2054, 2065–2697, ≥2697 kcal), family history of diabetes (yes or no), hypertension (yes or no), alanine aminotransferase (ALT, high or normal), gamma-glutamyl transferase (GGT, high or normal) and estimated glomerular filtration rate (eGFR). PIR is the ratio of the family’s self-reported income to the family’s appropriate poverty threshold according to the U.S. Census Bureau, and PIR values of 1.00 or greater indicate people above the poverty threshold [24]. The alcohol consumption (as obtained from dietary record) was categorized by 2 drinks/day for men and 1 drink/day for women. To calculate daily energy intake, participants reported the food and beverage items consumed between midnight and midnight 24 h prior to the NHANES dietary interview [25]. High ALT was defined as ALT > 43 units/L for men or >31 units/L for women, and high GGT was defined as GGT > 58 units/L for men or >35 units/L for women. We derived a dichotomous variable to indicate whether or not the participant met the 2018 US National Physical Activity Guidelines of ≥150 min of moderate activity or ≥75 min of vigorous activity per week, or an equivalent combination [26]. The serum creatinine-based CKD-epidemiology collaboration (CKD-EPI) equation was used to estimate eGFR [27].

### 2.5. Statistical Analysis

All data are expressed as mean (SD), median (IQR) or count (percentages). Student’s *t*-test, Chi-square or analysis of variance (ANOVA) were used for group comparisons. Urinary metals were divided by urinary creatinine to control the concentration dilution of urine. All metal concentrations were log10-transformed for association analysis to reduce skewness. We performed three stages of data analyses to identify the main and joint associations in urinary metal mixtures that had the most substantial influence on T2D risk (Appendix A).

In stage-1, we first applied the Bayesian kernel machine regression (BKMR) model to estimate the overall effects of the mixture of 15 metals on the prevalence of T2D [17]. We used the elastic net penalty regression to select the most important individual metals that were associated with T2D [28,29]. The elastic net model can perform selection while enabling the inclusion of collinear predictors through combining the least absolute shrinkage and selection operator and ridge [30]. We performed 10-fold cross-validation to acquire an unbiased and robust estimate of prediction accuracy.

In stage-2, we applied the BKMR model to eight metals selected in stage-1 to estimate their overall effects on T2D. Posterior inclusion probabilities (PIPs) were used to estimate the importance of each individual metal. We also applied logistic regression models to examine the associations of urinary metals and the odds of T2D adjusting for sex, age, race/ethnicity, educational attainment, PIR, smoking status, alcohol consumption, physical activity, BMI, average daily energy intake, family history of diabetes, hypertension, ALT, GGT, and eGFR. Linear trends across increasing quartiles (Q) of urinary metal concentrations were tested by assigning the median levels in quartiles and treating them as continuous variables [31]. We further estimated the associations between per log-10 increment of urinary metals with prevalence of T2D. We used variance inflation factor (VIF) to estimate collinearity in eight urinary metals in the logistic regression models [32]. Considering the limitation of BKMR models in estimating the effects of co-exposure patterns with both high and low levels of metals, we additionally performed weighted quantile sum (WQS) regression to compare the urinary metals identified as most important in the prevalence of T2D [33]. As WQS regression performs inference in a single direction (positive or negative direction) at a time, we separately applied the WQS model with both directions based on the effect direction of each metal from the elastic net regression.

In stage-3, we repeated the use of the BKMR models for the four most important metals identified in stage-2 by BKMR and the logistic regression models (Co, Sn, Sr and U). We further estimated the overall effect of four selected metals and their single dose–response associations as well as joint effects on T2D using both the BKMR model and logistic regression models.

We compared the overall effects of mixtures of between 15, 8, and 4 urinary metals as identified in three stages to evaluate whether the effects changed substantially by the number of included metals. We used both restricted cubic spline analysis with 3-knot (25th, 50th and 75th percentiles) and exposure–outcome function plots of the BKMR model to detect the shape of dose–response relationships of 4 metals identified in stage-3 with the odds for T2D [34]. In addition, we calculated the additive interactions for every two-metal pair expressed as relative excess risk due to interaction (RERI) and attributable proportion (AP) in the logistic regression model to estimate metal–metal interactions in 4 final identified metals [35].

Given the inherent nature of multiple complex survey designs, we accounted for the survey weights for each participant in the NHANES datasets in all models. In the logistic regression model, we used *svydesign* function in *R* accounting for sampling weights, as well as the stratification and clustering. All analyses were implemented using *R* version 4.0.3 software (R Foundation for Statistical Computing, Vienna, Austria). A two-sided *p* value of <0.05 was considered statistically significant.

## 3. Results

### 3.1. Characteristics of the Participants

There were 495 participants with T2D (12.4%) among the 3078 participants (Table 1). The mean ± SD age was 46.4 ± 17.0 years (58.2 ± 13.6 in participants with T2D and 44.1 ± 16.6 in those without T2D). The proportion of men and women was 54.1% and 45.9%, respectively. Except for PIR, there were statistical differences in the basic characteristics between participants with and without T2D. Participants with T2D were more likely to be non-Hispanic White, less educated, consume alcohol, and have hypertension. Most urinary metal concentrations were higher in participants with T2D than in participants without T2D, except for Ba, Mn, Hg, Sr, and TL (Table 1 and Appendix A).

### 3.2. Multi-Metal Selections

In the elastic net penalty regression model for identifying metals associated with the odds of T2D, we selected 8 out of 15 metals with non-zero coefficients of β (0.87 for Cd, −0.70 for Sr, 0.56 for Sn, −0.34 for Mn, 0.33 for Co, 0.23 for Mo, 0.18 for U, and −0.17 for TL) (Appendix A).

### 3.3. Single Urinary Metal Levels and the Risk of T2D

Figure 1 shows the adjusted ORs of T2D for urinary metals in the logistic regression models and their collinearity as well as the importance in BMKR models. Urinary Co (OR = 1.83, 95% CI = 1.00 to 3.34), Sn (OR = 1.53, 95% CI = 1.11 to 2.20), and U (OR = 1.39, 95% CI = 1.03 to 1.88) were positively associated with the prevalence of T2D, while an inverse association was found for urinary Sr (OR = 0.39, 95% CI = 0.23 to 0.64) (Appendix A). When looking into the collinearity issue of regression models between urinary metals with T2D (Figure 1), all VIF values were under 10, implying no substantial issue in collinearity. In the BKMR model, Co, Sr, Sn, and U had the highest PIPs among these eight urinary metals. The highest weight was also observed in the WQS model (weight: 0.34 for Co and 0.32 for Sn in the positive partial effect, and 0.50 of Sr in negative effect). The overall OR of T2D was 1.05 (95% CI: 1.01–1.08) for the positive effects of five metals and 1.00 (95% CI: 0.98–1.02) for the negative effects of three metals (Figure 2).

### 3.4. Dose–Response Associations and Their Interactions in Four Identified Metals

In the BKMR model, increased urinary Co, Sn, and U concentrations and decreased urinary Sr concentrations were linearly associated with increased prevalence of T2D (Appendix A). When examined by restricted cubic spline analysis with 3-knot, a positive linear dose–response relationship was observed for Co (*P*_overall_ = 0.008, *P*_non-linear_ = 0.100), and Sn (*P*_overall_ = 0.011, *P*_non-linear_ = 0.138), as well as an inverse linear relationship for Sr (*P*_overall_ = 0.001, *P*_non-linear_ = 0.209) (Figure 3).

The interaction analyses showed both positive additive (RERI = 0.57, 95% C.I. = 0.18 to 0.96; AP = 0.48, 95% C.I. = 0.16 to 0.80) and multiplicative (*P*_multi-interaction_ = 0.01) interactions between urinary Sn and Sr with T2D. There was no additive or multiplicative interaction between other metal–metal pairs with T2D (Table 2 and Appendix A). Comparing to participants with Sn ≥ 4.57 * 10^−2^ µg/g of Cr (median value) and Sr < 9.53 * 10^−2^ µg/g of Cr (median value), the lowest odds of T2D were observed in those with urinary Sn < 4.57 * 10^−2^ µg/g of Cr and Sr ≥ 9.53 * 10^−2^ µg/g of Cr (OR = 0.65, 95% C.I. = 0.45 to 0.92) (Appendix A).

### 3.5. Comparing Overall Effects of the Combined Urinary Metal Mixture on T2D

When comparing overall effects of metal mixtures between the 15, 8, and 4 urinary metals with the prevalence of T2D identified in 3 stages, the mixtures of four identified metal had similar overall effects compared with other combined metal mixtures in this study (Appendix A).

## 4. Discussion

In this comprehensive analysis with multiple novel statistical methods, we systematically identified the main and individual urinary metals as well as their interactions and the dose–response relationships of the most important single metal with the prevalence of T2D. Urinary Co, Sn, U and Sr had relatively higher importance to the prevalence of T2D when compared to other metals included in this study in U.S. adults. Urinary Co, Sn and U were positively associated with the odds of T2D, while urinary Sr was inversely associated with T2D. Sn and Sr also had significant interactive effects on the odds of T2D, where the levels of Sn modified the association between urinary Sr and T2D.

Estimation and quantification of the health impacts of environmental pollutants in epidemiological studies are a challenge. The conventional analysis of the relationship between metal exposure and cardiometabolic health while simultaneously adjusting models for several metals or introducing interaction terms may overlook the complexity in relationships within metal mixtures [17]. To increase the robustness of our findings, we performed an elastic net analysis and identified metals that may interact with diabetes and cardiometabolic status, including Cd, Co, Mn, Mo, Sr, Sn, TL, and U. Furthermore, we compared results from logistic regressions and BKMR models in multiple stages of analysis strategy and identified four metals (Co, Sn, U and Sr) that have higher relative importance to the prevalence of T2D in U.S. adults. We also quantified the relative importance of these four metals to the odds of T2D after accounting for inter-metal interactions, and quantified the importance in PIP values [36]. As demonstrated in PIP statistics in this study, Co, Sn, Sr and U might play more important roles in the odds of T2D. This finding agrees with our logistic regression analysis, where they were the only metals that demonstrated significant associations in linear models. Another interesting observation is that the overall effect of metal mixtures on the odds of T2D did not differ substantially regardless of the number of included metals, which suggests that the four metals identified by this study may play the most important roles in the risk of T2D.

In this analysis, urinary Co, Sn, Sr and U had independent associations with the prevalence of T2D, a finding that was supported by evidence from laboratory and population studies [3,12,37]. Animal studies found that tributyltins (chemical compounds based on Sn with hydrocarbon substituents) could impair pancreatic function [38,39], induce insulin resistance [40] and disrupt glucose homeostasis [41]. In NHANES 2011–2014, urinary Sn concentrations were higher in individuals with diabetes [42]. As for U, the chemical toxicity is a much greater concern than radioactivity in its naturally occurring form, and its association with T2D was also significant in NHANES 1999–2010 [13].

Moreover, Co is an essential element in the human body and is a component of vitamin B12 [43]. The glycemia-lowering effect of cobalt chloride decreased systemic glucose production, increased tissue glucose uptake, or induced a combination of the two mechanisms. Co could cause an increased expression of glucose transporter 1 (GLUT1) and inhibition of gluconeogenesis [44]. Interestingly, we observed a positive association between elevated urinary Co and prevalence of T2D. Previous population-based studies on the association between Co and T2D have not been consistent, and the results might differ according to the types of specimens [45,46]. Additionally, exposure to heavy metals may negatively affect kidney function, which may bias the associations between urinary metals and prevalence of diabetes in cross-sectional studies. The discrepancies in observations do warrant further prospective cohort studies.

Although not considered an essential trace element, the role of Sr in endocrine metabolism is worthy of attention [47]. Our study found an inverse association between urinary Sr and T2D from logistic regression and a dose–response effect as revealed by restricted cubic spline analysis, which was consistent with previous in vivo and vitro studies [48,49]. Studies found that Sr has potential anti-diabetic effects in terms of antioxidant [50] and fat metabolism [51]. Sr can regulate the expression of pancreas- and kidney-related genes in diabetic mice by glucose regulation and improve insulin tolerance [47,49]. Although knowledge about the role of Sr in T2D development is limited [52], one added value of our study was the evaluation of the interactive effects of urinary Sr and Sn on their association with T2D. The significant protective effect of Sr relative to the odds of T2D was attenuated by a higher level of Sn. Although a mechanistic explanation in the context of physiology is still lacking, the impairment of glucose metabolism may offset the antioxidative effect of Sr.

The strength of the present study is the use of data collected by rigorous protocols and extensive quality-control procedures in the NHANES survey. The conventional statistical methods are limited by multiple comparisons testing, multicollinearity and potential model misspecifications to handle high-dimensional and correlated exposures as well as to assess their interactions [53]. Using both conventional and multiple advanced machine learning-based statistical models, we systematically evaluated the main effects of metal mixtures and identified the most important single-metal and combinations of metal mixtures as well as their dose–response relationships and interactions by more sophisticated statical approaches. However, the findings should be interpreted carefully by noting several limitations. First, residual confounding effects could not be fully addressed, such as the influence of metals from dietary sources and other unmeasured risk factors. In addition, due to its cross-sectional nature, the present study was not able to rule out how changes in lifestyle factors or medication use may affect the levels of urinary metals and cardiometabolic biomarkers. Another limitation was the measurement of metal exposure at single time points, which may not reflect long-term and cumulative exposure. Moreover, findings from our study sample may not be generalized to the total population in the U.S. despite accounting for the sampling weight. Finally, since metal mixtures interact with each other in the human body, the interactions and dose–response associations among important metals identified in this study did not account for other metals, which could cause biases, and did not provide the overall interactions within the metal mixtures.

To conclude, urinary Co, Sn, U and Sr were important metals in the prevalence of diabetes in U.S. adults. The levels of Sn might modify the effect of Sr on T2D. More mechanistic studies and prospective cohorts are needed to verify the dose–response associations and joint effects in the onset of diabetes and other cardiometabolic diseases.

## Figures and Tables

**Figure 1 nutrients-14-01552-f001:**
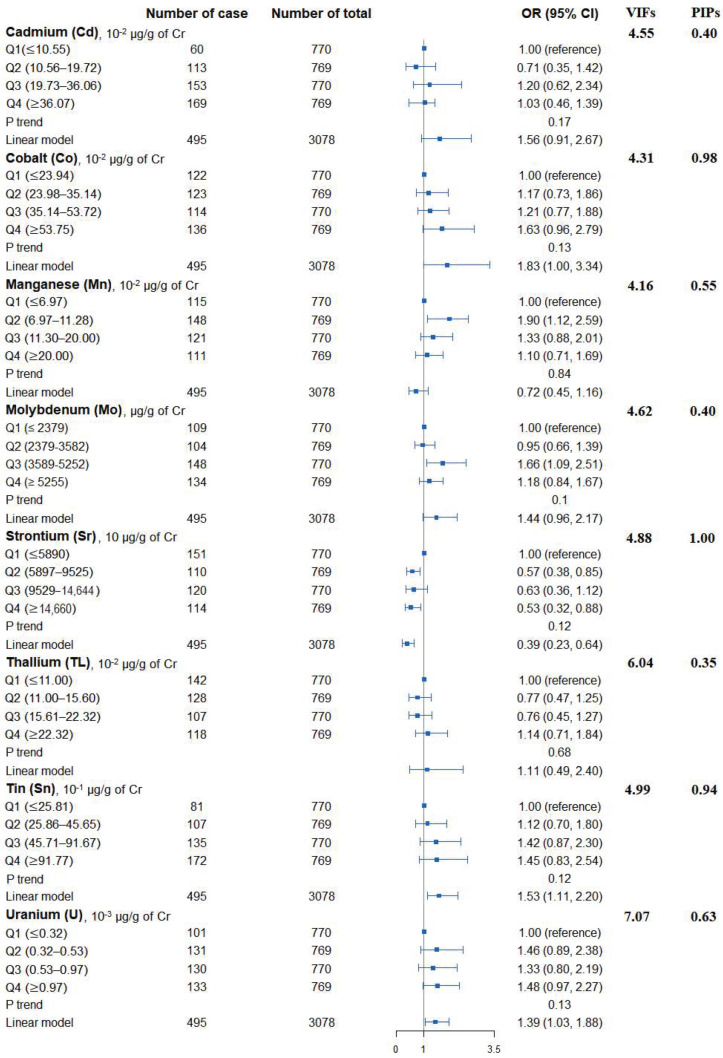
Associations of urinary metals with prevalence of type 2 diabetes, their collinearity in logistic regression models, and the importance in BKMR models. Model adjusted for sex, age, race/ethnicity, educational attainment, poverty-income ratio, smoking status, alcohol consumption, physical activity, body mass index, average daily energy intake, family history of diabetes, hypertension, alanine aminotransferase, gamma-glutamyl transferase, and estimated glomerular filtration rate.

**Figure 2 nutrients-14-01552-f002:**
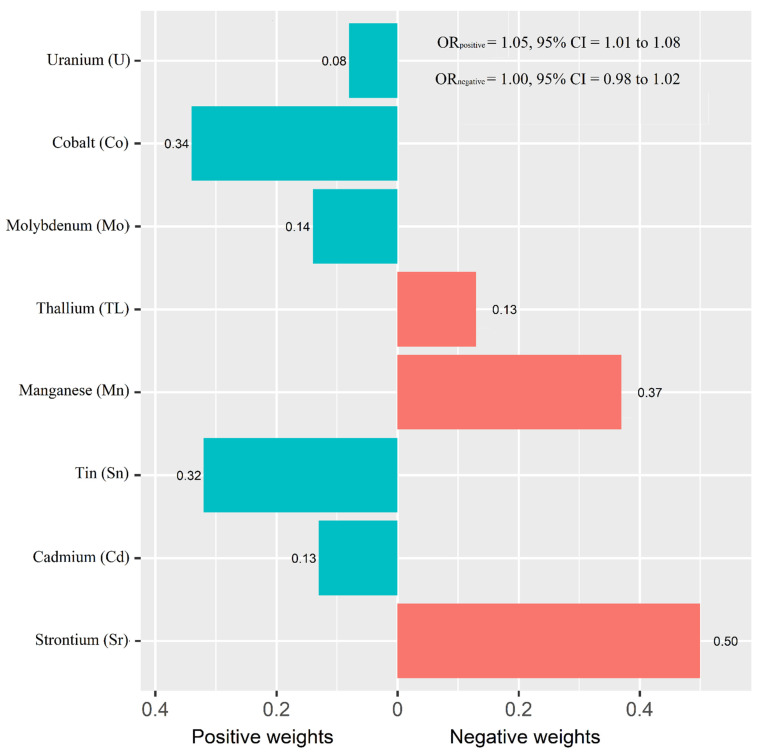
Weights representing the proportion of the positive or negative partial effect for each metal in the weighted quantile sum (WQS) regression. Model adjusted for sex, age, race/ethnicity, educational attainment, poverty-income ratio, smoking status, alcohol consumption, physical activity, body mass index, average daily energy intake, family history of diabetes, hypertension, alanine aminotransferase, gamma-glutamyl transferase, and estimated glomerular filtration rate.

**Figure 3 nutrients-14-01552-f003:**
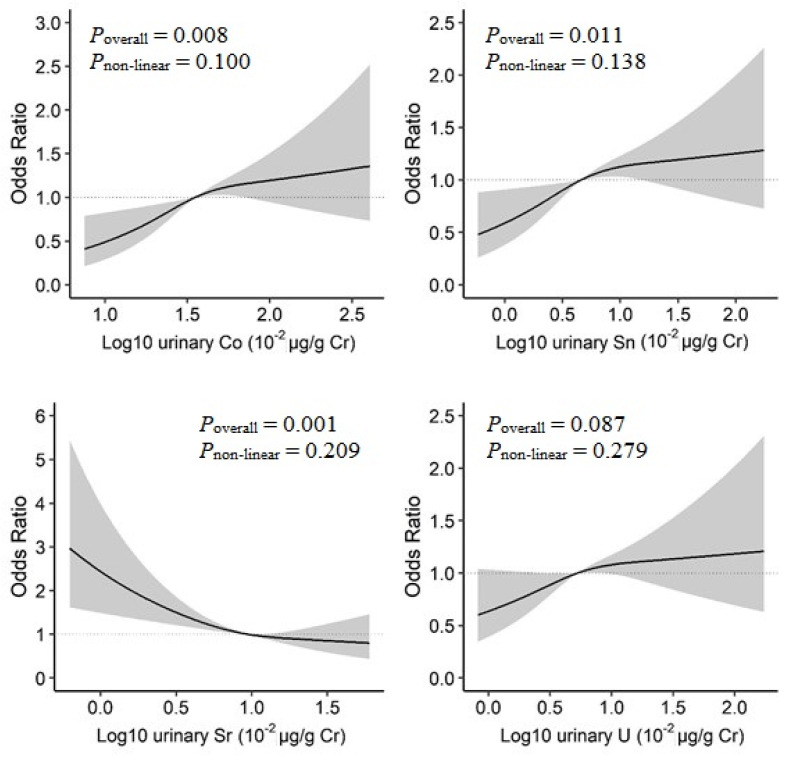
Dose–response relationships of urinary metals and prevalence of type 2 diabetes in the restricted cubic spline analysis. Model adjusted for sex, age, race/ethnicity, educational attainment, poverty-income ratio, smoking status, alcohol consumption, physical activity, body mass index, average daily energy intake, family history of diabetes, hypertension, alanine aminotransferase, gamma-glutamyl transferase, and estimated glomerular filtration rate.

**Table 1 nutrients-14-01552-t001:** Basic characteristics of participants by type 2 diabetes (T2D) in NHANES 2011–2016.

Variables ^a^	Non-T2D (*n* = 2583)	With T2D (*n* = 495)	Total (*n* = 3078)	*p*-Value
Survey cycle				0.002
2011–2012	782 (30.3)	156 (31.5)	938 (30.5)	
2013–2014	963 (37.3)	146 (29.5)	1109 (36.0)	
2015–2016	838 (32.4)	193 (39.0)	1031 (33.5)	
Men	1363 (52.8)	302 (61.0)	1665 (54.1)	<0.001
Age, years old	44.1 ± 16.6	58.2 ± 13.6	46.4 ± 17.0	<0.001
18–39	1181 (45.7)	54 (10.9)	1235 (40.1)	
40–59	854 (33.1)	184 (37.2)	1038 (33.7)	
≥60	548 (21.2)	257 (51.9)	805 (26.2)	
Race				0.038
Non-Hispanic White	1112 (43.1)	186 (37.6)	1298 (42.2)	
Non-Hispanic Black	528 (20.4)	109 (22.0)	637 (20.7)	
Hispanics	589 (22.8)	138 (27.9)	727 (23.6)	
Other race	354 (13.7)	62 (12.5)	416 (13.5)	
Education				<0.001
Less than high school	411 (15.9)	117 (23.6)	528 (17.2)	
High school	529 (20.5)	110 (22.2)	639 (20.8)	
At least some college	1643 (63.6)	268 (54.1)	1911 (62.1)	
Poverty income ratio				0.684
Below poverty (<1)	627 (24.3)	125 (25.3)	752 (24.4)	
At or above poverty (≥1)	1956 (75.7)	370 (74.7)	2326 (75.6)	
BMI, kg/m^2^				<0.001
≤25	860 (33.3)	70 (14.1)	930 (30.2)	
25.1–29.9	866 (33.5)	143 (28.9)	1009 (32.8)	
≥30	857 (33.2)	282 (57.0)	1139 (37.0)	
Smoking status				<0.001
Never smoker	1392 (53.9)	222 (44.8)	1614 (52.4)	
Former smoker	600 (23.2)	172 (34.7)	772 (25.1)	
Current smoker	591 (22.9)	101 (20.4)	692 (22.5)	
Alcohol consumption				<0.001
No	1352 (52.3)	209 (42.2)	1561 (50.7)	
Yes	1231 (47.7)	286 (57.8)	1517 (49.3)	
Physical activity				0.034
Substandard	2208 (85.5)	441 (89.1)	2649 (86.1)	
Standard	375 (14.5)	54 (10.9)	429 (13.9)	
Average daily energy intake				0.004
Q1 (<1515 kcal)	519 (20.1)	117 (23.6)	636 (20.7)	
Q2 (1515–2054 kcal)	536 (20.8)	120 (24.2)	656 (21.3)	
Q3 (2065–2697 kcal)	631 (24.4)	126 (25.5)	757 (24.6)	
Q4 (≥2697 kcal)	897 (34.7)	132 (26.7)	1029 (33.4)	
With hypertension	849 (32.9)	346 (69.9)	1195 (38.8)	<0.001
Family history of diabetes	566 (21.9)	54 (10.9)	620 (20.1)	<0.001
ALT				<0.001
Normal	2262 (87.6)	401 (81.0)	2663 (86.5)	
High	321 (12.4)	94 (19.0)	415 (13.5)	
GGT				<0.001
Normal	2342 (90.7)	410 (82.8)	2752 (89.4)	
High	241 (9.3)	85 (17.2)	326 (10.6)	
eGFR (ml/min per 1.73 m^2^)	143 ± 12.2	141 ± 15.4	142 ± 12.8	<0.001
Urinary metal concentrations				
Antimony (Sb), 10^−2^ µg/g of Cr	4.83 (3.31, 7.44)	4.97 (3.56, 7.12)	4.85 (3.35, 7.27)	<0.001
Arsenic (As), µg/g of Cr	6.77 (3.96, 14.37)	7.52 (4.39, 14.34)	6.86 (4.02, 14.35)	<0.001
Barium (Ba), µg/g of Cr	1.15 (0.65, 2.07)	1.03 (0.51, 2.23)	1.13 (0.62, 2.08)	<0.001
Cadmium (Cd), µg/g of Cr	0.19 (9.85, 34.20)	0.26 (0.16, 0.44)	0.20 (0.12, 0.36)	<0.001
Cesium (Cs), µg/g of Cr	4.12 (3.03, 5.85)	4.41 (3.10, 5.82)	4.15 (3.03, 5.85)	<0.001
Cobalt (Co), µg/g of Cr	0.35 (0.24, 0.53)	0.36 (0.24, 0.56)	0.35 (0.24, 0.54)	<0.001
Lead (Pb), µg/g of Cr	0.35 (0.22, 0.58)	0.40 (0.25, 0.60)	0.36 (0.22, 0.58)	<0.001
Manganese (Mn), 10^−1^ µg/g of Cr	1.14 (0.69, 2.00)	1.06 (0.70, 1.85)	1.13 (0.70, 2.00)	<0.001
Mercury (Hg), µg/g of Cr	0.28 (0.14, 0.58)	0.27 (0.13, 0.56)	0.28 (0.14, 0.58)	<0.001
Molybdenum (Mo), µg/g of Cr	35.18 (23.41, 52.22)	39.08 (25.71, 53.64)	35.86 (23.79, 52.52)	<0.001
Strontium (Sr), 10 µg/g of Cr	9.60 (6.01, 14.78)	9.03 (4.83, 13.92)	9.53 (5.89, 14.66)	<0.001
Thallium (Tl), µg/g of Cr	0.16 (0.1, 0.22)	0.15 (0.10, 0.22)	0.16 (0.11, 0.22)	<0.001
Tin (Sn), µg/g of Cr	0.43 (0.25, 0.85)	0.59 (0.34, 1.28)	0.46 (0.26, 0.92)	<0.001
Tungsten (W), 10^−2^ µg/g of Cr	5.93 (3.51, 10.59)	6.19 (3.71, 10.28)	5.97 (3.53, 10.53)	<0.001
Uranium (U), 10^−3^ µg/g of Cr	5.21 (3.13, 9.62)	5.67 (3.44, 10.20)	5.26 (3.19, 9.71)	<0.001

Abbreviations: T2D, type 2 diabetes; BMI, body mass index; ALT, alanine aminotransferase; GGT, gamma-glutamyl transferase; eGFR, estimated glomerular filtration rate. ^a^ Values are presented as *n* (%), median (IQR), or mean ± SD.

**Table 2 nutrients-14-01552-t002:** Interactions between urinary metals and type 2 diabetes.

	RERI (Relative Excess Risk Due to Interaction)	AP (Attributable Proportion)	*P*_multi-interaction_ ^a^
Co and Sn	0.14 (−0.39, 0.67)	0.09 (−0.25, 0.43)	0.75
Co and Sr	−0.23 (−0.79, 0.33)	−0.21 (−0.73, 0.30)	0.45
Co and U	−0.02 (−0.53, 0.48)	−0.02 (−0.41, 0.38)	0.89
Sn and Sr	0.57 (0.18, 0.96) *	0.48 (0.16, 0.80) *	0.01 *
Sn and U	0.02 (−0.51, 0.55)	0.01 (−0.36, 0.38)	0.96
Sr and U	−0.02 (−0.48, 0.44)	−0.02 (−0.46, 0.41)	0.95

^a^ Model adjusted for sex, age, race/ethnicity, educational attainment, poverty-income ratio, smoking status, alcohol consumption, physical activity, body mass index, average daily energy intake, family history of diabetes, hypertension, alanine aminotransferase, gamma-glutamyl transferase, and estimated glomerular filtration rate. * *p* < 0.05.

## Data Availability

The data for this study will not be made publicly available because the NHANES database is publicly available and in the public domain.

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
