# Peer review of "Identifying Effects of Urinary Metals on Type 2 Diabetes in U.S. Adults: Cross-Sectional Analysis of National Health and Nutrition Examination Survey 2011–2016"

_nutrients, 2022, doi:10.3390/nu14081552_

Round 1

Reviewer 1 Report

The manuscript reviewed has a great value to strengthen the knowledge of the Environmental Epidemiology in areas where the exposure to mixed metals happens, also, in low levels. The methodological approach based on novel statistical methods could respond to research questions related to exposure to mixed chemicals and metabolic or other chronic conditions.

Congratulations! 

The manuscript reviewed has a great value to strengthen the knowledge of the Environmental Epidemiology in areas where the exposure to mixed metals happens, also, in low levels. The methodological approach based on novel statistical methods could respond to research questions
related to exposure to mixed chemicals and metabolic or other chronic conditions.

Comment 1:
In the Material and methods section (2.2 T2D definition), I suggest adding some details about the definition used for T2D; the authors do not report if were included participants that use medications for T2D.

Comment 2

In section 2.4: On covariates, please add more details on the mean to add covariates that could be correlated; how was defined estimated glomerular filtration rate?

Comment 3:

Line 192: Please verify the typing mistake in the word “RARE”.

Comment 4:

As a recommendation, it could be useful to add in the Section Discussion more details on the strengths and limitations due to the use of novel statistical methods. This experience could be useful in other contexts, especially in developing countries.

Comment 5:

For Table S1, consider presenting in the main manuscript a short descriptive table that shows the statistical descriptor for the metals evaluated. In Supplementary Material is useful to show all the details. Please review some types of mistakes.

Comment 6:

For Table S2, consider presenting in the main manuscript a short descriptive table that shows the adjusted Odds ratio for the best models for the prevalence of T2D by urinary metal levels

Author Response

Reviewer 1:

The manuscript reviewed has a great value to strengthen the knowledge of the Environmental Epidemiology in areas where the exposure to mixed metals happens, also, in low levels. The methodological approach based on novel statistical methods could respond to research questions related to exposure to mixed chemicals and metabolic or other chronic conditions. Congratulations! 

The manuscript reviewed has a great value to strengthen the knowledge of the Environmental Epidemiology in areas where the exposure to mixed metals happens, also, in low levels. The methodological approach based on novel statistical methods could respond to research questions related to exposure to mixed chemicals and metabolic or other chronic conditions.

Comment.1
In the Material and methods section (2.2 T2D definition), I suggest adding some details about the definition used for T2D; the authors do not report if were included participants that use medications for T2D.

Responses:

We appreciate the reviewers’ comments and suggestions on our manuscript. We also included participants who were self-reported taking insulin or oral glucose-lower drugs at baseline in the NAHANES survey. We have now clarified it in section of 2.2 T2D definition.

Comment 2
In section 2.4: On covariates, please add more details on the mean to add covariates that could be correlated; how was defined estimated glomerular filtration rate?

Responses:

We selected potential risk factors in this analysis based on the priori knowledge on the risk of diabetes. We have added more detailed description and references in section of 2.4 Covariables in the revised manuscript. We used the CKD-EPI (Chronic Kidney Disease Epidemiology Collaboration) equation to estimate estimated glomerular filtration rate (eGFR). We have now clarified it in 2.4 Covariables.

Comment 3:
Line 192: Please verify the typing mistake in the word “RARE”.

Responses:

We have corrected all typos accordingly. Thank you.

Comment 4:
As a recommendation, it could be useful to add in the Section Discussion more details on the strengths and limitations due to the use of novel statistical methods. This experience could be useful in other contexts, especially in developing countries.

Responses:

Thank you for the comments. Estimation and quantification of the health impacts of environmental pollutants in epidemiological studies is a challenge. The conventional statistical methods are limited by multiple comparisons testing, multicollinearity and potential model misspecifications to handle high-dimensional and correlated exposures as well as to assess their interactions. In this comprehensive analysis, we adopted both the conventional and multiple advanced machine learning-based statistical models. In the second paragraph of 4. Discussion in the revised manuscript, we have highlighted the strengths of this study with multiple advanced statistical methods and pointed out the potential limitations such as the interactions and dose-response associations among identified metals in this study did not account for other metals which could cause biases and did not provide the overall interactions in overall metal mixtures as pointed out by the second reviewer.

Comment 5:
For Table S1, consider presenting in the main manuscript a short descriptive table that shows the statistical descriptor for the metals evaluated. In Supplementary Material is useful to show all the details. Please review some types of mistakes.

Responses:

We have added distributions of urinary metal levels in Table 1 of the main text. We also checked typos and thoroughly revised and edited the main text.

Comment 6:
For Table S2, consider presenting in the main manuscript a short descriptive table that shows the adjusted Odds ratio for the best models for the prevalence of T2D by urinary metal levels.

Responses:

We have merged main results of Table S2 in the updated Figure 1 with the associations of urinary metals with prevalence of diabetes in logistic models, their collinearity, and the importance in BKMR models.  

Reviewer 2 Report

The authors conducted a study about " Identifying effects of urinary metals on type 2 diabetes in U.S. adults: Cross-sectional analysis of National Health and Nutrition Examination Survey 2011–2016". This is not a novel study, but the authors attempted to use advanced statistical methods to identify these relationships. Several parts of the manuscript need to be clarified, especially the method section. Please see details below.

  • Criteria for selecting confounders? As the authors mentioned, the association between heavy metals and T2DM is consistent because previous studies adjusted covariates differently. The authors select average daily energy intake, alcohol consumption, and smoking as confounders, while physical activity is missing? It has been known that physical activity is an important covariate when analyzing the effects of urinary heavy metals on human health effects. Furthermore, if the survey cycle is a significant contributor to T2DM, why did the authors not include it in this study?
  • Although authors used advanced statistics (BMKR is the key method) to identify the association between urinary heavy metals and T2DM, BKMR has limitations (limited in estimating the effects of co-exposure patterns with both high and low levels of chemicals). Therefore, I recommend authors use more methods to compare with the current methods and select the optimal heavy metals for the final analysis, such as weighted quantile sum (WQS) and gqcomp (Quantile G-Computation). However, since heavy metals interact with each other in the human body, if the authors select the important heavy metals to conclude that heavy metals are linked with T2DM, it will cause biases and not provide the whole picture of heavy metal interactions, especially since there is no substantial issue with collinearity in this study.
  • The authors only assess the association between heavy metals and T2DM (lack of its components). Thus, I recommend further analysis with prediabetes, glucose, and HbA1C. There are many papers assessing the mixed effects of heavy metals on health effects. The authors can refer to

References:

DOI: 10.1007/s11356-022-18871-2

https://doi.org/10.1016/j.intimp.2021.108428

  • BKMR allows for possible interactions and nonlinear relationships, but the authors did not provide this information. The readers did not know how these heavy metals interact.
  • In the method section, please provide how many samples have values under LODs. Because for analytic results below the lower LODs, the values are imputed as the lower limit of detection divided by the square root of two. to ensure that the readers can look at the exact distribution of metals.
  • Figure S1 did not provide detail information without metals measurements and covariates.
  • Figure S5 did not show 95%CI.

Author Response

Reviewer 2:

The authors conducted a study about " Identifying effects of urinary metals on type 2 diabetes in U.S. adults: Cross-sectional analysis of National Health and Nutrition Examination Survey 2011–2016". This is not a novel study, but the authors attempted to use advanced statistical methods to identify these relationships. Several parts of the manuscript need to be clarified, especially the method section. Please see details below.

Responses:

Thank you for these comments and suggestions.

Criteria for selecting confounders? As the authors mentioned, the association between heavy metals and T2DM is consistent because previous studies adjusted covariates differently. The authors select average daily energy intake, alcohol consumption, and smoking as confounders, while physical activity is missing? It has been known that physical activity is an important covariate when analyzing the effects of urinary heavy metals on human health effects. Furthermore, if the survey cycle is a significant contributor to T2DM, why did the authors not include it in this study?

Responses:

The covariates included in this analysis were based on the priori knowledge on the risk of diabetes. We have now added a more detailed description and references in the section of 2.4 Covariables in the revised manuscript. The NHANES collections use complex and multi-stage survey sampling to ensure that results are representative of the US population. In this analysis, we accounted for the survey sample weights for each participant in all models. In the logistic regression models, we used svydesign R function accounting for sampling weights as well as the stratification and clustering. We have now clarified this in the section of 2.5. Statistical analysis.

Although authors used advanced statistics (BMKR is the key method) to identify the association between urinary heavy metals and T2DM, BKMR has limitations (limited in estimating the effects of co-exposure patterns with both high and low levels of chemicals). Therefore, I recommend authors use more methods to compare with the current methods and select the optimal heavy metals for the final analysis, such as weighted quantile sum (WQS) and gqcomp (Quantile G-Computation). However, since heavy metals interact with each other in the human body, if the authors select the important heavy metals to conclude that heavy metals are linked with T2DM, it will cause biases and not provide the whole picture of heavy metal interactions, especially since there is no substantial issue with collinearity in this study.

Responses:

Thank you very much for the insightful suggestions. Inspired by the reviewer, we further performed WQS models to compare the identified metals by BKMR model. The WQS model also shows Co, Sr and Sn were the important metals in the mixtures. We have now added the related methods and results in 2.5. Statistical analysis and 3.3. Single urinary metal levels and the risk of T2D of the revised manuscript.

The authors only assess the association between heavy metals and T2DM (lack of its components). Thus, I recommend further analysis with prediabetes, glucose, and HbA1C. There are many papers assessing the mixed effects of heavy metals on health effects. The authors can refer to

References:

DOI: 10.1007/s11356-022-18871-2

https://doi.org/10.1016/j.intimp.2021.108428

Responses:

We appreciate these comments. In the current study, we try to focus on estimating and identifying the overall and individual effects of metal mixtures and their interactions with type 2 diabetes. We are also interested in exploring the associations of heavy metals with prediabetes, metabolic syndrome, fasting glucose, HbA1c, insulin functions and other cardiometabolic biomarkers as the following our publications with conventional statistical methods. We will adopt novel statistical methods in our future studies for providing more evidence on linking of heavy metals with risk diabetes and cardiometabolic diseases.

  1. Yang J, Yang A, Cheng N, et al. Sex-specific associations of blood and urinary manganese levels with glucose levels, insulin resistance and kidney function in US adults: National health and nutrition examination survey 2011-2016. Chemosphere. 2020;258:126940.
  2. Lo K, Yang JL, Chen CL, et al. Associations between blood and urinary manganese with metabolic syndrome and its components: Cross-sectional analysis of National Health and Nutrition Examination Survey 2011-2016. Sci Total Environ. 2021;780:146527.
  3. Yang A, Liu S, Cheng Z, et al. Dose-response analysis of environmental exposure to multiple metals and their joint effects with fasting plasma glucose among occupational workers. Chemosphere. 2017;186:314-321.

BKMR allows for possible interactions and nonlinear relationships, but the authors did not provide this information. The readers did not know how these heavy metals interact.

Responses:

We have now described this information in 3. Results, and also proved figures (Figure S4 and Figure S5) in supplementary materials.

In the method section, please provide how many samples have values under LODs. Because for analytic results below the lower LODs, the values are imputed as the lower limit of detection divided by the square root of two. to ensure that the readers can look at the exact distribution of metals.

Responses:

We have described the details of LODs in the main text and Table S1.

Figure S1 did not provide detail information without metals measurements and covariates.

Responses:

We have now updated and provided details in the updated Figure S1.

Figure S5 did not show 95%CI.

Responses:  

We have added these figures.

Round 2

Reviewer 2 Report

The authors conducted a study about " Identifying effects of urinary metals on type 2 diabetes in U.S. adults: Cross-sectional analysis of National Health and Nutrition Examination Survey 2011–2016".

Thanks for giving us a second chance to review this manuscript again. In this version, the authors all almost resolved the issues which were mentioned in the previous version. However, I have minor comments. In the abstract, the authors only reported the findings from BKMR and logistic regression model, without WQS model. It has been that the aims of this study are to assess the interactions between chemicals and T2DM. Logistic regression model can not assess their interactions, and have more limitations rather WQS. Thus, I recommend authors add more findings from WQS model in the abstract.

Author Response

Thank you for these comments. We have now added the main findings from WQS model in both abstract and main text of the manuscript.  Thank you.